# Bioconversion of Furanic Compounds by *Chlorella vulgaris*—Unveiling Biotechnological Potentials

**DOI:** 10.3390/microorganisms12061222

**Published:** 2024-06-18

**Authors:** Ricarda Kriechbaum, Oliver Spadiut, Julian Kopp

**Affiliations:** Research Division: Biochemical Engineering, Integrated Bioprocess Development, Institute of Chemical, Environmental and Bioscience Engineering, Technische Universität Wien, Gumpendorferstraße 1a, 1060 Wien, Austria; ricarda.kriechbaum@tuwien.ac.at (R.K.); oliver.spadiut@tuwien.ac.at (O.S.)

**Keywords:** 5-Hydroxymethylfurfural, furfural, conversion reaction, aldehyde oxidation, 5-Hydroxymethyl-2-Furoic Acid, 2-Furoic Acid, lignocellulosic hydrolysates, inhibitors

## Abstract

Lignocellulosic biomass is abundant on Earth, and there are multiple acidic pretreatment options to separate the cellulose, hemicellulose, and lignin fraction. By doing so, the fermentation inhibitors 5-Hydroxymethylfurfural (HMF) and furfural (FF) are produced in varying concentrations depending on the hydrolyzed substrate. In this study, the impact of these furanic compounds on *Chlorella vulgaris* growth and photosynthetic activity was analyzed. Both compounds led to a prolonged lag phase in *Chlorella vulgaris* growth. While the photosynthetic yield Y(II) was not significantly influenced in cultivations containing HMF, FF significantly reduced Y(II). The conversion of 5-Hydroxymethylfurfural and furfural to 5-Hydroxymethyl-2-Furoic Acid and 2-Furoic Acid was observed. In total, 100% of HMF and FF was converted in photoautotrophic and mixotrophic *Chlorella vulgaris* cultivations. The results demonstrate that *Chlorella vulgaris* is, as of now, the first known microalgal species converting furanic compounds.

## 1. Introduction

There is an abundance of lignocellulosic biomass on Earth, which is currently being utilized in the paper and pulp industry [1], bioethanol production [2], or varying biorefinery approaches [3]. In order to extract monomeric C5 and C6 carbon sources from the hemicellulose or cellulose fraction, these biomasses have to be pretreated. Treatments can be performed either enzymatically, which is very costly, or through thermo-acid hydrolysis [4]. One of the main disadvantages of the latter is the formation of microbial growth inhibitors, furfural (FF), 5-Hydroxymethylfurfural (HMF), acetate, or multiple different phenolic compounds in the acidic hydrolysis of polymeric carbohydrates [5]. FF stems from the dehydration of pentoses [6], while HMF originates from hexoses [7], resulting in varying furan ratios depending on the lignocellulosic biomass hydrolyzed. Furanic compounds are known to damage or alter DNA structures [1,8], damage cell walls and membranes [6], and inhibit growth and enzymatic activities [9] in multiple different microorganisms.

Detoxification of so-called fermentation inhibitors, HMF, FF, and phenolic compounds is crucial to designing feasible processes, as yields and microbial growth are retarded by their presence. Detoxification processes can include lime addition [8], activated carbon resins [10], distillation [11,12], or organic solvent extraction [13]. However, chemical and mechanical strategies often lead to losses in monomeric sugar content and are energy- and time-intensive. Therefore, bioconversions would be a feasible tool for the detoxification of furanic compounds. Increasing research activities are currently focused on the implementation of microbial processes, including native pathways [1,14,15] of furan detoxification or generating recombinant strains [16,17], expressing identified enzymes isolated from strains that do possess these abilities.

Table 1 depicts multiple microorganisms able to metabolize HMF and FF into different furan metabolites. Enzymes mentioned are either of native origin (^N^) or recombinantly produced (^R^) by the mentioned microorganisms. Enzymes responsible for furfural detoxification are described more often in the literature than HMF due to furfural’s higher toxicity and the ability to increase the toxic potential of other compounds [8].

Recently, it was shown that wheat straw hydrolysates from liquid hot water pretreatment composed of monomeric and oligomeric sugars, acetic acid, and furanic compounds can be used for *Chlorella vulgaris* cultivation [22]. During those studies, the depletion of furanic compounds in the growth medium was observed. There have been studies investigating the potential of microalgal growth on lignocellulosic hydrolysates [23,24] or wastewater containing furfural [25,26], but to the author’s knowledge, conversion of furanic compounds in microalgae was never reported.

This study investigated said depletion and identification of the fate of 5-Hydroxymethylfurfural and furfural during microalgal cultivation. For this purpose, phototrophic cultivations were set up with varying concentrations of HMF and FF, as well as mixotrophic cultivation with additional glucose, to further investigate the potential of using waste streams resulting from acid-hydrolyzed biomasses. Potential metabolites were identified, and the conversion rates and yields of those reactions were calculated. The quantum yield of the photosystem II (PSII) was measured to identify any influence of HMF and FF on photosynthesis. Additionally, three synthetic hydrolysates mimicking the composition of actual hydrolysates obtained from liquid hot water pretreatment were used as substrates to test the interrelation of the HMF/FF ratio within those hydrolysates on *Chlorella vulgaris*.

## 2. Materials and Methods

### 2.1. Microalgae Strain and Inoculum Preparation

*Chlorella vulgaris* (UTEX 2714), obtained from the Culture Collection of Algae at the University of Texas at Austin, was used in this study. BG11 [22] at pH 7.5 was used as the cultivation medium. The inoculum was cultivated at 23 °C, with 3% (*v*/*v*) CO_2_-enriched air, shaking at 100 rpm in a Minitron shaker (Infors HT, Bottmingen, Switzerland). Illumination was performed in cycles, with consecutive 14 h light (100 µmol/m^2^/s) and 10 h darkness phases.

### 2.2. Shakeflask Experiments

*Chlorella vulgaris* was cultivated at consecutive 14 h light (100 µmol/m^2^/s) and 10 h darkness cycles at 23 °C in a 3.0% (*v*/*v*) with CO_2_-enriched air in a Minitron shaker (Infors HT, Basel, Switzerland). An amount of 30 mL of BG11 pH 7.5 containing either 400 mg/L of HMF or FF was added sterilely to a 100 mL shakeflask before inoculation to a starting optical density at 600 nm (OD_600_) of 0.10. To account for changes in mixotrophic cultivation mode, shakeflasks were prepared as described above with an additional 1 g/L glucose. Shakeflasks without microalgal inoculum were prepared and sampled accordingly to check for furanic compound reduction due to volatile properties. Sampling was performed three times per week for 3 weeks. One mL aliquots were centrifuged at 10,000× *g*, and 4 °C for 10 min, and the supernatant was used for high-performance liquid chromatography and high-performance ion chromatography to determine furan and anion content during cultivation. Additionally, 200 µL aliquots were taken to determine the influence on photosynthetic activity F_V_/F_M_ during the cultivation with furanic compounds. 

### 2.3. Mimicking 5-Hydroxymethylfurfural and Furfural Composition of Hydrolysates

To determine the applicability of *Chlorella vulgaris* cultivation with hydrolysates obtained from biorefinery approaches, three hydrolysates from the liquid hot water pretreatment of lignocellulosic biomass were mimicked in terms of carbon and furan content (Table 2). The dried feedstock used for the liquid hot water (LHW) hydrolyzation in this work were wheat straw [27], apple cores, and banana peels. Triplicates of the LHW pretreatment were carried out in a stainless-steel high-pressure autoclave (Zirbus, HAD 9/16, Bad Grund, Germany). Approximately 30 g of dried biomass was dissolved in 330 g MQ-H_2_O. The hydrolyzation was carried out at 160 °C for 90 min. Solids and liquids were then separated using a hydraulic press (Hapa, HPH 2.5, Achern, Germany) and the liquid fraction was analyzed for sugar content and furanic compounds (Section 2.7). The concentrations of HMF and furfural of the three different lignocellulosic biomass hydrolysates were normalized to the monomeric glucose content for the screening experiments.

These normalized substrates were prepared according to Table 2 and sterile filtered (0.22 µm) before inoculation with *Chlorella vulgaris* at an initial optical density at 600 nm—OD_600_ of 0.1. Screening experiments with those mimicked hydrolysates were carried out in biological triplicates and sampled according to the first shakeflask experiment.

### 2.4. Determination of Growth Performance and Conversion

Cell growth was determined by measuring OD_600_ on a Nanodrop One photometer (Thermo Fisher Scientific, Waltham, MA, USA). To determine the correlation between dry cell weight—DCW—and OD_600_ measurements, dried biomasses were weighed at varying corresponding optical densities at 600 nm. Equation (1) displays the correlation. Growth rates and substrate uptake rates were calculated according to Equations (2) and (3).
DCW = 0.283 × OD_600_(1)
(2)$$\mu \;=\frac{(\ln{\mathrm{DCW}}_{2}-\ln{\mathrm{DCW}}_{1})}{t_{2}-t_{1}}$$
(3)$$q_{s}=\frac{1}{{\mathrm{DCW}}_{2}}\times \frac{S_{2}-S_{1}}{t_{2}-t_{1}}$$

OD_600_ optical density at 600 nm; µ: growth rate (day^−1^); DCW_x_: dry cell weight (g/L) at timepoint t_x_ (day); q_s_: specific substrate uptake rate (g/g/day); S_x_: substrate concentration (g/L) at timepoint t_x_.

5-Hydroxymethylfurfural (HMF) and furfural (FF) conversion and yield Y_F/F_ (%) were calculated according to Equations (4) and (5). Furfural evaporation was measured and added to the conversion and yield calculation accordingly. HMF did not evaporate during cultivation.
(4)$$\mathrm{Conversion}\;=\frac{c_{\mathrm{Furan}\;\mathrm{consumed}}}{c_{\mathrm{Furan}\;\mathrm{added}}}\;\times \;100\%$$
(5)$$Y_{F/F}\;=\frac{c_{\mathrm{Furan}\;\mathrm{metabolite}\;\mathrm{produced}}}{c_{\mathrm{Furan}\;\mathrm{added}}}\;\times \;100\%$$

c_Furan consumed_: furanic compounds metabolized—HMF or FF (mg/L); c_Furan added_: furanic compounds added—HMF or furfural (mg/L); Y_F/F_: furan metabolite yield from furanic compounds (%); c_Furan metabolite produced_: furan metabolites (HMFA or FA) produced (mg/L).

### 2.5. Detection of Metabolites—LC-IMS-QTOFMS

The supernatants of the aliquoted samples were analyzed using an Agilent 1290 Infinity II UPLC (Agilent Technologies, Santa Clara, CA, USA), equipped with a diode array detector (Agilent Technologies, Santa Clara, CA, USA) set to 277 nm and 285 nm. In addition, all spectra were stored as well in the range of 200 to 400 nm. A 250 mm × 4.6 mm Zorbax Extend-C18 column (Agilent Technologies, Santa Clara, CA, USA) was used at 25 °C with a flow rate of 0.3 mL/min. The method describing the gradient was published elsewhere [28]. In serial configuration, the flow was directly linked to the Series 6560 Ion Mobility—QTOFMS (Agilent Technologies, Santa Clara, CA, USA) mass spectrometer equipped with a Jetstream ESI source in positive ion, DDA mode (MS-range: 50–1700 *m*/*z*; DDA = switching to MS/MS mode for eluting peaks; selection range: 50–1700 *m*/*z*). Instrument calibration was performed using ESI calibration mixture (Agilent Technologies, Santa Clara, CA, USA). The instrument was tuned in fragile ion mode. Data were processed using MassHunter Workstation Qualitative Analysis Version 10.0 (Agilent Technologies, Santa Clara, CA, USA).

### 2.6. High-Performance Liquid Chromatography

The aliquoted supernatants of the samples were analyzed for glucose and furanic compounds using a Vanquish Core system (Thermo Fisher Scientific, Waltham, MA, USA). It was equipped with a variable wavelength diode array detector (Thermo Fisher Scientific, Waltham, MA, USA) at a wavelength of 235 nm, 277 nm, and 285 nm and a Refractomax 521 refractive index detector (Thermo Fisher Scientific, Waltham, MA, USA). An Aminex HPX-87H was used as an analytical column at 60 °C with 4 mM H_2_SO_4_ as the mobile phase at a flow velocity of 0.6 mL/min, running for 50 min isocratically. Quantification was performed by measuring standards of pure substances (Carl Roth, Karlsruhe, Germany) and establishing a calibration curve. Controlling, monitoring, and evaluation of the analysis were performed with Chromeleon 7.2.10 Chromatography Data System (Thermo Fisher Scientific, Waltham, MA, USA).

### 2.7. High-Performance Ion Chromatography

The aliquoted supernatants of the samples were also analyzed for anion metabolization. A Dionex Integrion HPIC System (Thermo Fisher Scientific, Waltham, MA, USA) equipped with a conductivity detector unit (Thermo Fisher Scientific, Waltham, MA, USA), combined with an ADRS 600 suppressor (Thermo Fisher Scientific, Waltham, MA, USA) and a CR-TC continuously regenerated trap column (Thermo Fisher Scientific, Waltham, MA, USA) was used. An IonPac AS11-HC column, including a guard column, was used as a stationary phase at 30 °C. The mobile phase was 100% ultrapure water generating a gradient with an EGC 500 KOH eluent generator cartridge (Thermo Fisher Scientific, Waltham, MA, USA). The binary gradient started at 0.2 mM and was raised to 5 mM KOH in 2.4 min. In 12.6 min, a linear gradient to 24 mM was applied, then increased to 38 mM in the following 17.4 min. Afterwards, there was an isocratic hold step for 5.1 min, followed by a linear decrease to 0.2 mM in 2.5 min. There was an isocratic equilibration step for 10 min at 0.2 mM KOH to finish. The flow velocity was 0.3 mL/min. Quantification was performed by measuring standards of pure substances (Carl Roth, Karlsruhe, Germany) and establishing a calibration curve. Controlling, monitoring, and evaluation of the analysis were performed with Chromeleon 7.2.10 Chromatography Data System (Thermo Fisher Scientific, Waltham, MA, USA).

### 2.8. Photosynthetic Activity

The photosynthetic activity of the cultures was monitored using a WATER-PAM-II chlorophyll fluorometer (Walz, Effeltrich, Germany) using pulse-amplitude modulation fluorometry. The software used was WinControl-3. The maximum photochemical efficiency F_V_/F_M_ of the photosystem II and the photochemical quantum yield Y(II) was calculated. The samples were diluted 40-fold with 4 mL ultrapure H_2_O (Milli-Q, Merck, Darmstadt, Germany) and adapted to darkness for 20 min in a dark space prior to measurement to ensure that all reaction centers of PSII were closed.

### 2.9. Statistical Analysis

Experiments were carried out in biological triplicates. Averages and standard deviations are reported. Statistical analysis was performed using one-way ANOVA (post hoc Tukey test) in Origin Pro 2021b (OriginLab Corporation, Northampton, MA, USA) with a *p*-value of <0.05 considered significant.

## 3. Results and Discussion

### 3.1. Screening Experiments

During a previous study, depletion of HMF and FF in *Chlorella vulgaris* cultivation in hydrolysates obtained from liquid hot water pretreatment [22] was observed; therefore, further investigation into the conversion of furanic compounds was needed. In order to determine furan metabolism in microalgae, screening experiments were performed photoautotrophically and mixotrophically with the addition of glucose. In Figure 1, *Chlorella vulgaris* growth DCW (g/L) and the corresponding photosynthetic yields Y(II) can be seen. The influence of FF on *Chlorella vulgaris* growth (Figure 1a) and photosynthetic yield (Figure 1c) was more pronounced than the influence of HMF on *C. vulgaris* growth (Figure 1b) and photosynthetic yield Y(II) (Figure 1d).

During *Chlorella vulgaris* cultivation in BG11, a two-day long lag phase could be seen, after which exponential growth started. In mixotrophic cultivation with the addition of glucose, no lag phase was visible. When adding FF to the cultivation medium, the lag phase lasted for approximately 7 days (Figure 1a). Afterwards, conventional mixotrophic and photoautotrophic growth behavior could be obtained. This directly correlated with the decrease in photosynthetic activity until day 7 (Figure 1c) in FF cultivations. Y(II) describes the ratio of the number of electrons transported through the PSII compared to the number of photons absorbed during photosynthesis. The decrease in Y(II) could represent an inhibition in the photosynthetic electron transport chain and, therefore, reduced production of ATP and NADPH. This reduced NADPH production is already stated in the literature for other organisms [14]. Excitation of a chlorophyll molecule by a photon leads to the transfer of this excited energy to PSII. If all of the reaction centers were closed, this redundant excitation energy could lead to the formation of reactive oxygen species, especially singlet oxygen ^1^O_2_* [29].

The addition of HMF to photoautotrophic and mixotrophic *C. vulgaris* cultivations did not result in this extremely prolonged lag phase (Figure 1b), hinting that HMF was not as toxic to the cells as FF. The photosynthetic yield Y(II) was also not significantly influenced in HMF-containing cultivations compared to photoautotrophic cultivations or mixotrophic cultivations in BG11. Therefore, HMF did not notably influence the photosynthetic machinery, whereas FF did. 

In Table 3, the specific growth rates µ (day^−1^) are shown, depicting the shifted growth curve of *Chlorella vulgaris* cultivated with the addition of FF. The growth rates increased to 0.63 day^−1^ in photoautotrophic cultivation and 0.76 day^−1^ in mixotrophic cultivation after FF was completely depleted (µ_(t=7–10)_). The effect of HMF on microalgal growth was not as prominent, where the growth rates were decreased by about 40% at t = 0–4 but adjusted after 4 days to catch up on the photoautotrophic control. In mixotrophic cultivation with HMF, the initial growth rate was similar to that of photoautotrophic cultivation with HMF, which was reduced by 40% as well. The growth rate was maintained over a longer period of time, as glucose was metabolized more slowly, compared to the mixotrophic control cultivation.

Studies with different microorganisms suggest that FF might be more toxic to metabolic machineries than HMF [15]. The presence of FF also increases the microbial toxicity of other compounds present after acid hydrolysis, like HMF [8]. The inhibitory effects of furans on microbial metabolisms are not fully understood yet, but researchers suggest that these compounds directly influence the enzymes needed for glycolysis and increase the formation of reactive oxygen species [14].

### 3.2. Influence on Nutrient Uptake

To gain insights into possible metabolic pathways in FF and HMF conversion, the nutrient uptake of the different media was compared. In Figure 2a–h, the uptake of FF, HMF, glucose, nitrate, and phosphate is depicted in the FF and HMF screening experiments. Furfural conversion takes 7 days, during which no glucose or nitrate uptake occurs (Figure 2c,e). This also explained the 7-day lag phase in mixotrophic and photoautotrophic cultivations (Figure 1a). Phosphate conversion was decreased by FF but not completely inhibited. Phosphate is needed for ATP and NADP generation and to fulfill basic microalgal energy needs, suggesting metabolism was slowed down, but not completely hindered during FF cultivation of *Chlorella vulgaris* (Figure 2g). While glucose was already depleted at the first sampling point in the mixotrophic control cultivation, glucose metabolization only started after 7 days in the experiments with FF present. Furfural conversion was not influenced by the presence of glucose at all, as both mixotrophic and phototrophic cultivation showed the same decrease in furfural concentration over time (Figure 2a). After FF was completely metabolized from the culture, Y(II) recovered to about 0.70, which is the ratio also measured in BG11 and BG11 + Glc *Chlorella vulgaris* cultivations. Therefore, no permanent damage, like chloroplast disruption or pigment degradation, was done to the PSII, but rather an inhibition in electron transfer took place, possibly accompanied by the generation of reactive oxygen species.

HMF in *Chlorella vulgaris* medium did not influence nutrient uptake as strongly as FF. HMF was depleted after 7 days in both mixotrophic and photoautotrophic cultivation (Figure 2b). Glucose uptake was decreased in the presence of HMF but not completely inhibited, as for FF (Figure 2d). Nitrate uptake, on the other hand, was slightly decreased in the presence of HMF in both the mixotrophic and photoautotrophic cultivation of *C. vulgaris* (Figure 2f). Phosphate uptake also slightly decreased in the presence of HMF (Figure 2h). Biodegradation of other VOCs, such as phenolic compounds, was preferably metabolized in the presence of organic carbon sources, such as glucose and acetate [30], but these correlations did not seem to be the case with either HMF or FF and glucose addition.

Overall, these results were in agreement with the literature that FF is more toxic to cells compared to HMF [8,31], as it completely retarded cell growth, decreased the photosynthetic quantum yield of the photosystem II Y(II), and retarded the glucose and nitrate uptake in *Chlorella vulgaris*.

### 3.3. Mimicking Hydrolysates

To test for possible interactions of HMF and FF present in the cultivation medium, *Chlorella vulgaris* was cultivated on three different synthetic hydrolysates mimicking the monomeric glucose, furfural, and HMF content of hydrolysates obtained from the liquid hot water (LHW) pretreatment of apple cores, banana peels, and wheat straw. The synthetic hydrolysate compositions were normalized to the monomeric glucose content of the respective actual hydrolysates. The three tested synthetic hydrolysates represent three different HMF/FF ratios, as they are decreasing from apple cores at 10.3 to banana peels at 4.7 and wheat straw at 0.4 (Table 2). In the synthetic wheat straw hydrolysate, there was the highest FF content at about 434 mg/L, which is the more toxic compound compared to HMF. This led to the longest lag phase of about 14 days (Figure 3a), as furfural became detoxified to furoic acid. The decrease in metabolic activity can also be seen in the photosynthetic quantum yield of photosystem II, which decreases up until day 14 and then slowly increases as furfural is completely metabolized (Figure 3b).

In all three synthetic hydrolysates, there was a visible lag phase before *Chlorella vulgaris* growth sets in. This lag phase was the most prominent in the synthetic wheat straw hydrolysate, as it lasted for 14 days. In these 14 days, the Y(II) significantly dropped, indicating a loss of photosynthetic function. But after complete furfural depletion, the Y(II) regenerated to its initial state. This phenomenon could be seen in all three synthetic hydrolysates. The depletion of the furanic compounds, glucose, nitrate, and phosphate is depicted in Figure 4. The timepoint of furfural conversion is depicted with the dotted line, which marks the spot where the metabolization of the other substrates was possible or the conversion of HMF began. 

This suggested that working with actual hydrolysates might be feasible, but the FF concentration and, additionally, the HMF/FF ratio have to be considered (Figure 3). The production of phenolic or aromatic compounds depends on the hydrolyzation parameters, such as temperature, acid concentration, process duration, and on the biomass chosen [32]. *Chlorella vulgaris* has been described to metabolize multiple different phenols and aromatics [33], but correlating interactions between furans and other compounds have to be assessed on a case-by-case basis. 

### 3.4. Identification of Converted Compounds

Comparing conversion products from *Chlorella vulgaris* to furan metabolites from known metabolic pathways in other organisms [1,4,14,15,16,19,21] using LC-QTOF-IM-MS, the products were identified as 5-Hydroxymethyl-2-Furoic Acid (HMFA) and furoic acid (FA) (Figure 5). Oxidation reactions to detoxify HMF and furfural are common in other microorganisms (Table 1), but the responsible enzymes vary. The identified metabolites HMFA and FA matched with commercially purchased standards (Merck, Darmstadt, Germany) in terms of retention time (19.90 min and 28.90 min) and their determined mass/charge ratio (143.0339 *m*/*z* and 113.9233 *m*/*z*).

A possible conversion reaction could be catalyzed by an enzyme of the oxidoreductase class in *Chlorella vulgaris* (Figure 6), as the aldehyde group on the furan ring becomes oxidized in both cases. While the genome of *Chlorella vulgaris* has been completely sequenced [34], not all genes have been functionally annotated. There are currently 422 oxidoreductases identified based on homology, 7 of those being aldehyde oxidases or aldehyde dehydrogenases [35].

However, BLASTing the genome of *Chlorella vulgaris* with amino acid sequences of enzymes of known furan oxidizing species did not show any significant or very low sequence similarities (Appendix A).

Studies on the microalgal production of volatile organic compounds (VOCs) have recently been increasing in the field of “Volatilomics” [36]. These compounds are described to be involved in microalgal communication and detoxification processes in liquid cultures [37]. Biodegradation of aromatic and volatile compounds has been described in the field of microalgal wastewater remediation. The degradation of benzoic acid [38], hydroxybenzoic acid [39], xylene [40], and ethylbenzene [41] has been described multiple times in microalgal cultivations. However, furans and their derivatives are only known in the microalgal context to be produced and released as volatile organic compounds into the environment [42,43].

There are two possibilities for the conversion of furanic compounds in *Chlorella vulgaris* cultivation: (i) Furanic compounds are known to induce the formation of reactive oxygen species in other organisms, such as Saccharomyces cerevisiae. Furans are also known to be oxidized by ^1^O_2_* [44] singlet oxygen, leading to the hypothesis that the production of reactive oxygen species in *Chlorella vulgaris* led to the oxidation of furans. However, ^1^O_2_* generated intracellularly in microalgal species has a very small reactive radius of 100 nm and typically does not travel far from PSII before reacting with a target molecule [29]. In addition, all microalgal species contain a well-working enzymatic and non-enzymatic machinery decreasing multiple reactive oxygen species intracellularly [45]. Intracellular ROS have also never been described as a means of oxidizing furanic compounds. (ii) *Chlorella vulgaris* is known to produce different furanic compounds (mainly furan rings bound to alkyl chains) allegedly as communication signals. This could indicate the availability of enzymes responsible for furanic detoxification, as the metabolic machinery for furan production is available. There are known enzymatic reactions in multiple different organisms responsible for the conversion of furfural to furoic acid and 5-Hydroxymethylfurfural to 5-Hydroxymethyl-2-Furoic Acid (Table 1), suggesting the likelihood of this hypothesis.

2-Furoic Acid and 5-Hydroxymethyl-2-Furoic Acid have been studied for their potential in industrial applications. 2-Furoic Acid has been used in optical applications [46], as resin materials and coatings [47] and in flavor and fragrance industries [48]. Whereas 5-Hydroxymethyl-2-Furoic Acid has been utilized as building blocks for bio-based polymers substituting for petrol-based polyethylene terephthalate [17,49,50,51,52]. These bio-based polymers could be used, for instance, as synthetic fibers or plastic bottles [17].

### 3.5. Conversion Rates and Yields

The calculated furan conversions (%) and the corresponding yield YF/F (%) are shown in Table 4. During photoautotrophic and mixotrophic cultivation, 100% of FF and HMF has been converted.

While 100% of FF and HMF was converted, the obtained yield Y_HMF/HMFA_ showed that 70–75% of HMF was converted to HMFA, whereas about 54% of FF was converted to FA (Y_FF/FA_) during the photoautotrophic and mixotrophic cultivation of *Chlorella vulgaris*. This suggests that the conversion of furanic compounds happens in multiple steps, and intermediates or by-products should still be identified and quantified to close the furan balance. In LC-QTOF-IM-MS analysis, no additional furan derivatives or intermediates from known metabolic pathways were identified. This could be due to the known volatility of several furanic compounds. Another possibility for the unclosed furan balance could be intermediates or by-products accumulating intracellularly, which might then be used by *Chlorella vulgaris* to generate signaling molecules based on furan rings, such as furan–methyl, furan–ethyl, and furan–pentyl [53].

## 4. Conclusions

This study demonstrates *Chlorella vulgaris* to be the first known microalgae species to convert furfural and 5-Hydroxymethylfurfural to 2-Furoic Acid and 5-Hydroxymethyl-2-Furoic Acid. 5-Hydroxymethyl-2-Furoic Acid could be further processed to bio-based polymers substituting for petrol-based plastics and 2-Furoic Acid might be applied in flavor and fragrance industries. This opens up a tremendous potential in regard to waste stream utilization, as these growth inhibitors are produced as by-products during the thermo-acidic hydrolyzation of C5 and C6 sugars. Hence, industrial waste streams containing furanic inhibitors are applicable for the production of -Furoic Acid and 5-Hydroxymethyl-2-Furoic Acid with *Chlorella vulgaris*. Even though this conversion with microalgae is unknown in the literature, further omics studies declaring the underlying mechanism have to be performed to identify one or multiple reasons for these oxidation reactions.

## Figures and Tables

**Figure 1 microorganisms-12-01222-f001:**
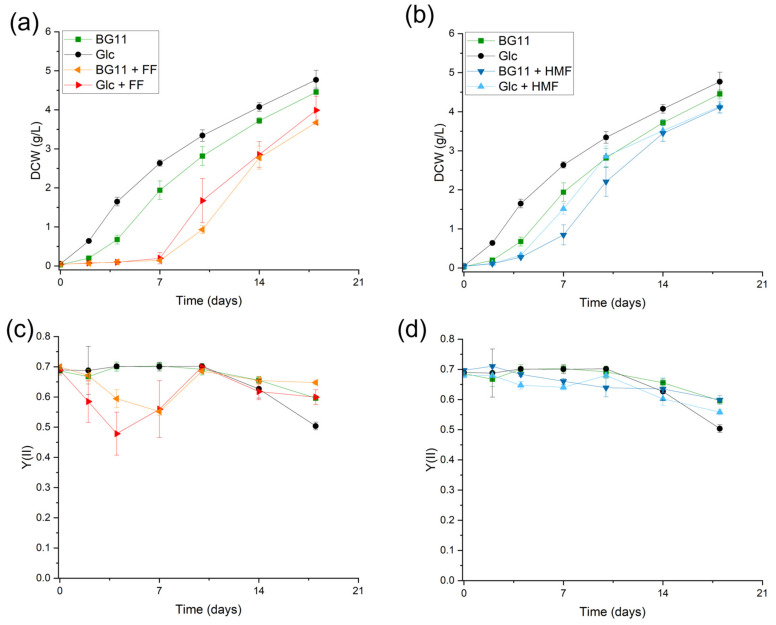
DCW (g/L) of *Chlorella vulgaris* grown on BG11, BG11 + Glc (**a**) BG11 + FF, BG11 + Glc + FF and (**b**) BG11 + HMF, BG11 + Glc + HMF and comparison of the effective quantum photosynthetic yield Y(II) of the different cultivations with either (**c**), FF or (**d**) HMF mixotrophic or photoautotrophic; HMF—5-Hydroxymethylfurfural; FF—furfural; Glc—glucose; DCW—dry cell weight (g/L); error bars represent standard deviations (n = 3).

**Figure 2 microorganisms-12-01222-f002:**
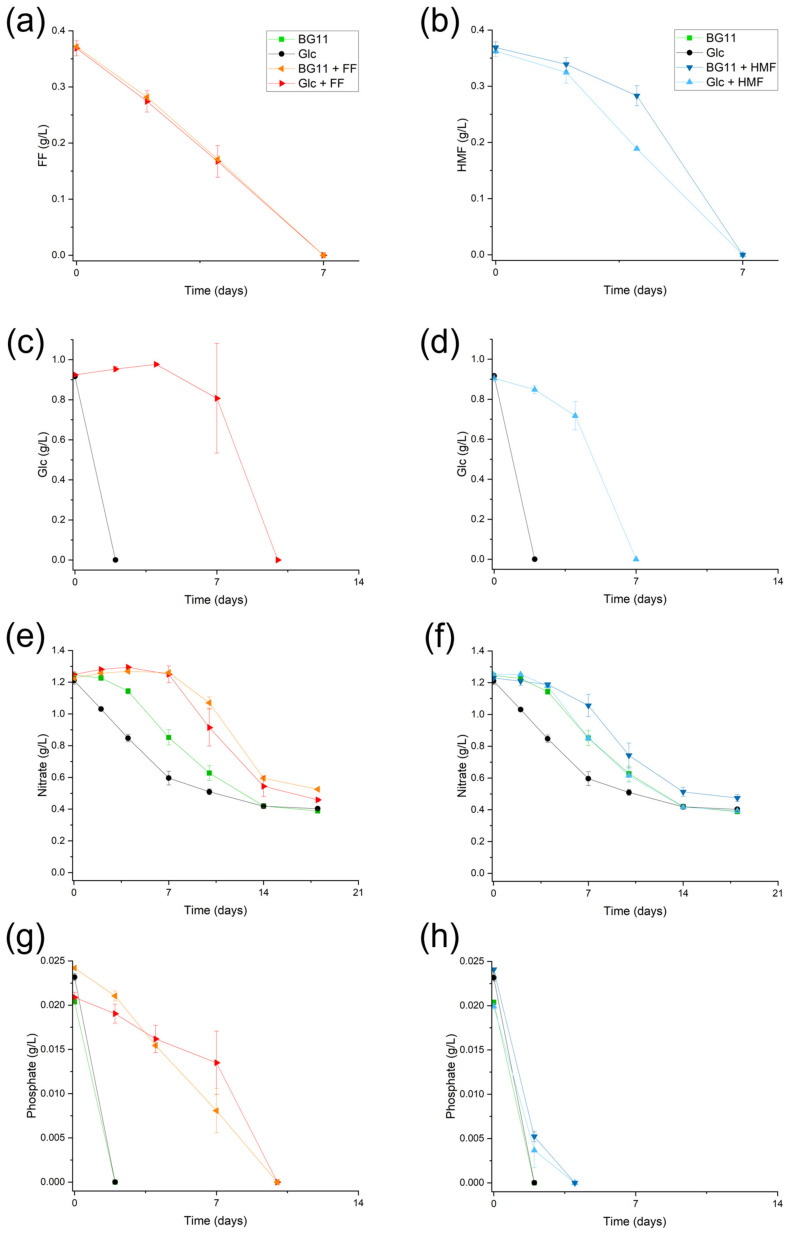
Substrate concentration (g/L) over time of (**a**) FF, (**c**) Glc, (**e**) nitrate, and (**g**) phosphate in mixotrophic and photoautotrophic experiments with furfural and (**b**) HMF, (**d**) Glc, (**f**) nitrate, and (**h**) phosphate in mixotrophic and photoautotrophic experiments with HMF; furfural—FF; 5-Hydroxymethylfurfural—HMF; glucose—Glc; error bars represent standard deviation (n = 3).

**Figure 3 microorganisms-12-01222-f003:**
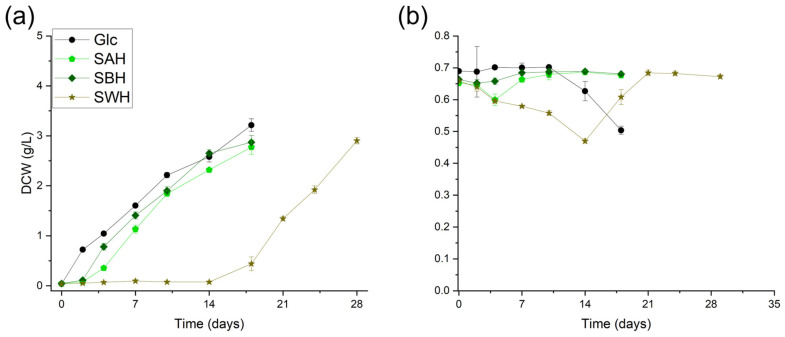
(**a**) DCW (g/L) of *Chlorella vulgaris* grown on synthetic hydrolysates, based on actual hydrolysates obtained from liquid hot water pretreatment of apple cores (SAH), banana peels (SBH), and wheat straw (SWH) compared to *Chlorella vulgaris* grown on glucose (Glc); DCW—dry cell weight (g/L), (**b**) comparison of the effective quantum photosynthetic yield Y(II) of the different cultivations on synthetic hydrolysates; error bars represent standard deviation (n = 3).

**Figure 4 microorganisms-12-01222-f004:**
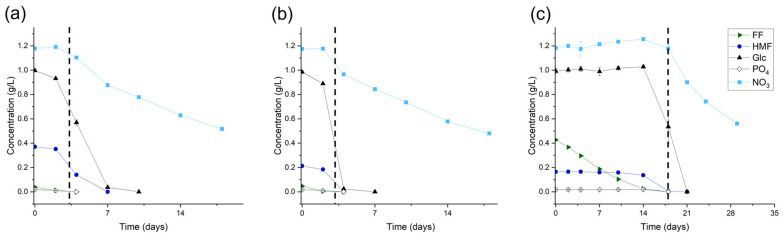
Substrate concentration (g/L) over time in the three synthetic hydrolysates, (**a**) apple cores, (**b**) banana peels, and (**c**) wheat straw. The dotted line marks the timepoint of complete furfural depletion; furfural—FF; 5-Hydroxymethylfurfural—HMF; glucose—Glc; phosphate—PO_4;_ nitrate—NO_3_; error bars represent standard deviation (n = 3).

**Figure 5 microorganisms-12-01222-f005:**
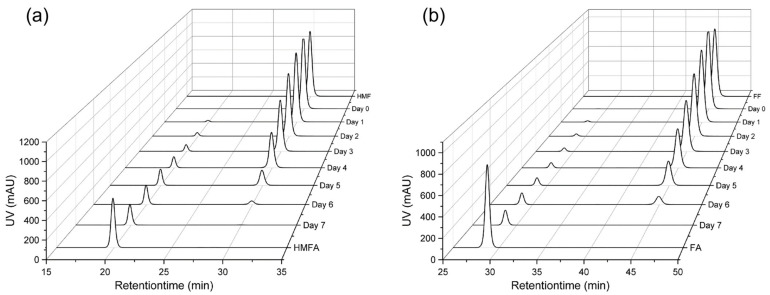
Conversion of (**a**) 5-Hydroxymethlyfurfurfal—HMF to 5-Hydroxymethyl-2-Furoic Acid—HMFA and (**b**) furfural—FF to furoic acid—FA during *Chlorella vulgaris* cultivation. Retention times and *m*/*z* ratios were compared to commercially available standards of the substances.

**Figure 6 microorganisms-12-01222-f006:**
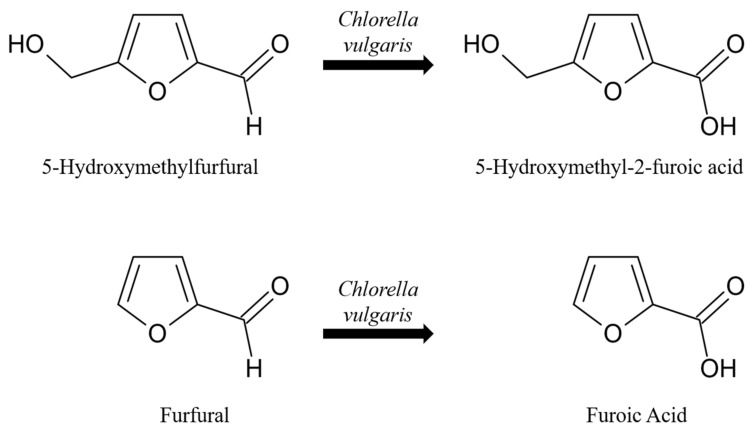
Identified reactions of 5-Hydroxymethylfurfural and furfural to 5-Hydroxymethyl-2-Furoic Acid and furoic acid during *Chlorella vulgaris* cultivation.

**Table 1 microorganisms-12-01222-t001:** Organisms capable of metabolizing 5-Hydroxymethylfurfural (HMF) and furfural (FF), the corresponding resulting metabolites and enzymes responsible for the conversion; ^R^—describes a recombinant enzyme, ^N^—describes a host native enzyme.

Organism	Furan	Metabolite	Responsible Enzymes	Reference
*Clostridium beijerinckii*	HMF	5-Hydroxymethylfurfuryl alcohol	Aldo/Keto Reductase ^N^	[15]
*Clostridium acetobutylicum*	HMF	2,5-bis-hydroxymethylfuran	Aldo/Keto Reductase ^N^	[6]
*Pseudomonas putida*	HMF	5-Hydroxymethyl-2-furoic acid	HMF/furfural oxidoreductase ^N^	[14]
*Escherichia coli*	HMF	5-Hydroxymethylfurfuryl alcohol	NADH-dependent aryl alcohol dehydrogenase (XylB) ^R^	[16]
		5-Hydroxymethyl-2-furoic acid	FAD-containing oxidase ^R^	[18]
		5-Hydroxymethyl-2-furoic acid	Vanillin dehydrogenase (CtVDH1) ^R^	[17]
*Cupriavidus basilensis*	HMF	2,5-Furandicarboxylic Acid	Furfural/HMF oxidoreductase (HMfABCDEFGH) ^N^	[19]
*Acinetobacter baylyi*	FF	Difurfuryl-Ether	Alcohol dehydrogenase (AreB) ^N^ Formaldehyde dehydrogenase (FrmA) ^N^	[4]
*Acinetobacter schindleri*	FF	Difurfuryl-Ether	Alcohol dehydrogenase (AreB) ^N^	[4]
*Cupridavidus necator*	FF	Furfuryl Alcohol	Zn-dependent alcohol dehydrogenase (FurX) ^N^	[20]
*Corynebacterium glutamicum*	FF	Furfuryl Alcohol	Furfural detoxification protein (FudC) ^N^	[1]
*Saccharomyces cerevisiae*	FF	Furfuryl Alcohol	NADH-dependent alcohol dehydrogenase (AHD1) ^R^	[21]
*Escherichia coli*	FF	Furoic Acid	Vanillin dehydrogenase (CtVDH1) ^R^	[17]
		Furoic Acid	FAD-containing oxidase ^R^	[18]
		Furfuryl Alcohol	NADH-dependent aryl alcohol dehydrogenase (XylB) ^R^	[16]
		Furfuryl Alcohol	NAPDH-dependent aldehyde reductase (YqhD) ^R^	[8]
*Clostridium beijerinckii*	FF	Furfuryl Alcohol	Aldo/Keto Reductase ^N^ Short-chain dehydrogenase/reductase ^N^	[15]
*Clostridium acetobutylicum*	FF	Furfuryl Alcohol	Aldo/Keto Reductase ^N^	[6]
*Saccharomyces carlsbergensis*	FF	Furoic Acid/Furfuryl Alcohol	n.a.	[2]
*Pseudomondas putida*	FF	Furoic Acid	HMF/furfural oxidoreductase ^N^	[14]
*Candida magnoliae*	FF	Furoic Acid	Aldehyde dehydrogenase ^N^	[9]
*Cupriavidus basilensis*	FF	2-oxoglutaric acid	Furfural/HMF oxidoreductase (HMfABCDEFGH) ^N^	[19]

**Table 2 microorganisms-12-01222-t002:** Comparison of the glucose—Glc, 5-Hydroxymethylfurfural—HMF, and furfural—FF content of different hydrolysates from varying lignocellulosic biomasses (wheat straw, apple cores, and banana peels) pretreated with liquid hot water (160 °C—90 min); normalization of HMF and FF content was based on glucose concentration.

Sample	Glc (mg/L)	HMF (mg/L)	FF (mg/L)	HMF/FF Ratio	Treatment
Wheat Straw	212	35	92	0.4	Original
	1000	165	434		Normalized
Apple Core	11,307	4207	410	10.3	Original
	1000	372	36		Normalized
Banana Peels	4289	910	195	4.7	Original
	1000	213	46		Normalized

**Table 3 microorganisms-12-01222-t003:** Comparison of *Chlorella vulgaris* growth rates µ (day^−1^), before and after furan conversion in photoautotrophic and mixotrophic cultivation. Errors represent standard deviation (n = 3).

(Day^−1^)	Photoautotrophic Cultivation			Mixotrophic Cultivation		
	BG11 (Control)	HMF	FF	BG11 + Glc	HMF	FF
µ_(t=0–4)_	0.78 ± 0.04	0.47 ± 0.07	0.23 ± 0.03	0.86 ± 0.04	0.51 ± 0.04	0.20 ± 0.04
µ_(t=4–7)_	0.35 ± 0.02	0.36 ± 0.07	0.10 ± 0.07	0.16 ± 0.03	0.51 ± 0.02	0.18 ± 0.21
µ_(t=7–10)_	0.13 ± 0.01	0.33 ± 0.05	0.63 ± 0.05	0.08 ± 0.01	0.21 ± 0.04	0.76 ± 0.14

**Table 4 microorganisms-12-01222-t004:** Conversion (%) and yields Y_F/F_ (%) of the furanic compounds in photoautotrophic and mixotrophic cultivation in *Chlorella vulgaris* cultivation; yields describing furanic compound conversion are describing either HMF to HMFA in Y_HMF/HMFA_ or FF to FA in Y_FF/FA_; HMF—5-Hydroxymethylfurfural; HMFA—5-Hydroxymethyl-2-Furoic Acid; FF—furfural; FA—2-Furoic Acid; error bars indicate standard deviation (n = 3).

(%)	Photoautotrophic		Mixotrophic	
	HMF	FF	HMF	FF
Conversion	100 ± 0	100 ± 0	100 ± 0	100 ± 0
Y_F/F_	76.35 ± 2.13	53.96 ± 1.93	69.68 ± 1.05	54.72 ± 3.66

## Data Availability

The raw data supporting the conclusions of this article will be made available by the authors on request.

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
