# Peer review of "Bioconversion of Furanic Compounds by Chlorella vulgaris—Unveiling Biotechnological Potentials"

_microorganisms, 2024, doi:10.3390/microorganisms12061222_

Round 1

Reviewer 1 Report

Comments and Suggestions for Authors

The reviewed MS is dedicated to describing new scientific data about the possibility of Chlorella vulgaris to convert furanic compounds. In the MS, mostly recent publications are cited. The experimental design is appropriate and based on modern methods of algae cultivation, biochemical, physiological, and statistical analysis. All parts of the paper are described in detail. The results are supported by tables and figures, the quality of some of the figures should be improved. The conclusions are consistent and based on results.

The MS can be recommended for publication after some corrections.

Major comments:

  1. Keywords: exclude Chlorella vulgaris, because you have mentioned it in the title. Add more keywords.
  2. Why did you use BG-11 media for Chlorella vulgaris cultivation? This media usually applied to cyanobacteria. Add an explanation to the section Materials and Methods.
  3. Mark the growth curves of chlorella and other parameters in Figures 1–4 by various colors, since it is difficult to distinguish curves in some diagrams.

Minor comments:

  1. Line 75: Please add at least 1-2 sentences about algae inoculum preparation. It is very inconvenient to find this information in previously published papers.
  2. Please enlarge Figure 1 to improve its visibility.
  3. Figure 6: Please improve the resolution of the figure and move it to the center.
  4. Correct the margins and spacing according to the journal template.

Author Response

We thank the reviewer for the comments provided and answered them point-by-point in the attached manuscript.

Reviewer 2 Report

Comments and Suggestions for Authors

I evaluated the mansucript entitled as: Chlorella vulgaris – the first known green algae to convert furanic compounds.

The manuscript is interesting but I reccomend major revisions before reconsider it for publication. Main concerns:

1) The title is not adequate. A lot of recent literature use lignocellulosic liquos to cultivate microalgae. Please, change it to reflect what the article really studied. 

For example: 

https://www.sciencedirect.com/science/article/abs/pii/S004896972402031X

https://link.springer.com/article/10.1007/s10311-022-01481-2

https://www.mdpi.com/1996-1073/7/7/4446

2) In the abstract section, more results should be presented. 

3) I did not understand the results where glucose was inserted in the medium and the microalga was cultivated phtoautotrophically. If an organic compound is present, is mixotrophycally. In several parts of the results and discussion, this is presented.

4) Pages 9-10, there is a blank page, please, adequate the mansucript to avoid it. 

5) In the introduction section, Table 1 needs to be converted in text. It is not the standard for original articles. 

6) Table 4 is not clear. Please revise this part and the section to be clearer to the readers. 

7) Section 3.4, other microorganisms able to convert these compunds need to mentioned. 

8) Section 2.3 need to be improved to describe sufficiently the lignocellulosic liquor with references and methodology. 

9) Sections in materials and methods have a lot of space beteween each other, pelase, take care about the mansucript presentations.

10) Fgures can be bigger to facilitate visualization. 

11) Equations need to be better presented. 

12) Please, pay attention to the discussion of the results and improve them. 

13) Conclusion section need to be improved. 

14) I saw that the furanic compounds were converted but not metabolized inside the microalgae. The results of this conversion, these new compounds have industrial applications. Put one paragraph regarding this at the end of the results and discussion section. 

Comments on the Quality of English Language

English is good but can be improved.

Author Response

(The authors gave the same response as above.)

Round 2

Reviewer 2 Report

Comments and Suggestions for Authors

I evaluated the revised manuscript.

The previous comments were addressed as required. I believe that in the current form, the paper can be accepted.